# Privatization in Rural Water Supply and Customer Satisfaction: An Empirical Case Study in Vietnam

**Nguyen Tuan Anh [1,*], Nguyen Huu Dung [2] and Dao Thi Thu [1]**

[1] Institute for Water Resources Economics and Management, Vietnam Academy for Water Resources, Dong Da District, Hanoi 11515, Vietnam; daothu.vkttl@gmail.com

[2] Faculty of Real Estate and Natural Resources Economics, National Economics University, Hai Ba Trung District, Hanoi 11616, Vietnam; nguyen.huudung@neu.edu.vn

* Correspondence: anhnt.iwem@gmail.com

**Abstract:** This article investigates the private sector participation in investment, management, and operation in rural water supply schemes in Vietnam. Different organizations manage rural water supply facilities, including the private sector, public sector, and others. This paper aims to compare the different characteristics affecting user satisfaction of water supply facilities managed by the private sector and the remaining sectors. An ordered logit model was utilized for calculation with the data collected from semi-structure questionnaires with 1200 households using water from rural water supply systems managed by different sectors in Vietnam. The results indicate that the water-user satisfaction with rural water supply projects managed by the private sector is higher than that in other sectors (community, cooperative, commune people's committee), whereas there is no significant difference in customer satisfaction between systems managed by the public sector and the private sector (enterprise, private management). The water availability and quality of schemes greatly influence the customer satisfaction. Findings from this study provide considerable information for the private sector on how to improve the management and operation of water supply systems efficiently through customer satisfaction assessment.

**Keywords:** customer satisfaction; privatization; private sector participation; rural water supply; water user

## 1. Introduction

The private sector plays a vital role in the investment, management, and operation of water supply schemes in many countries [1,2]. The private sector participation helps to mitigate the capital constraint for infrastructure development while improving the sustainability of exploiting water supply systems [3–5]. The private sector participation (PSP) brings many benefits such as the balance of socio-economic development, effective risk sharing, cost savings, efficient project implementation, technological innovation, and generate more investments in water supply infrastructure [6–8]. In addition, the private sector participates in the management and operation of water supply projects to improve operational efficiency and labor productivity [9–12], save costs and reduce water loss rate [11,13]. The involvement of public–private partnership (PPP) provides good water quality, expanded water supply networks, and higher levels of water-user satisfaction than before the implementation of policies involving PPP [14]. PSP is an effective approach to solve the challenges of water supply in rural areas and small towns [15,16]. Shifting rural water supply works from the public sector to the private sector is increasingly common due to the rapid population growth in small towns and rural commercial centers. This trend is becoming increasingly popular in Africa, especially in French-speaking countries such as Benin, Burkina Faso, Mali, Nigeria, Rwanda and Senegal [7,15], and in European countries Greece, Austria and Spain [7]. A similar pattern can also be seen in many European countries as they are expanding PSP by developing a comprehensive policy framework and consider

PSP as the primary tool to address issues related to infrastructure [17]. For instance, the proportion of PSP in investment, management, and operation of water supply systems in France has increased from 17% to 80% during the period of 1939–2001 [18]. Privatization of rural water supply is one of the essential decisions of the government to provide water supply services in rural areas, especially small towns. Preliminary evidence indicated an improvement in private sector-managed facilities' financial and operational indicators [19]. However, PSP also has some clear disadvantages, such as loss of control for state management agencies over water supply systems and higher price for consumers [20,21]. In some cases, the PSP did not bring significant improvements in efficiency [22] and did not offer significant improvements in the indicators of performance or customer satisfaction [23–26]. These contradictory claims warrant further investigation into the issue of PSP in water supply management.

Before the year 1999, most people used low-cost and straightforward water supply systems such as rainwater, water tanks, and wells in Vietnam. Since 2000, the Government of Vietnam has implemented a national strategy on rural water supply and environmental sanitation under Decision No. 104/2000/QD-TTg. Rural water and sanitation program was implemented in three phases between the years 2000–2005, 2006–2010, and 2011–2015. After 2016, the rural water supply and sanitation program has been integrated into the national target program to build new rural areas. After two decades, around 16,573 centralized water supply schemes have been built to supply water to 88.5% of the population and plays an important role in enhancing living conditions in rural areas [27]. In Vietnam, there are five different management models of water supply works, including (1) community, (2) cooperative, (3) enterprise, private; (4) commune people's committee, (5) provincial centre for rural water supply and environmental sanitation. In these five types, it can be divided into three management sectors, including (1) private sector (enterprise, private management); (2) public sector (provincial centre for rural water supply and Environmental Sanitation); (3) other sectors (community, cooperative, commune people's committee) [28].

In implementing the national target program on water and sanitation, the Government of Vietnam has issued several policies to attract the private sector participation such as decision no. 131/2009/QD-TT dated November 2, 2009, namely "several key preferential policies, encouraging investment, management, and exploitation of rural water supply work." Rural water supply projects with the private sector participation are entitled to preferential policies on land, tax, or support for initial investment costs. Many provinces of Vietnam have implemented incentive policies, provinces that have decided to implement preferential policies, the rate of "sustainable" operating works is 1.13 times higher than provinces without decided to implement preferential policies [29]. In addition, the government also issued Decree No. 57/2018/ND-CP in 2018 "mechanisms and policies to encourage enterprises to invest in agriculture and rural areas," which supports 3 million VND/m$^3$/day with capacity for new construction or 2 million VND/m$^3$/day incapacity for upgrading and renovating water supply factory and supporting up to 50% of water pipes line costs. The private sector is significantly involved in the management of the rural water supply, especially in Red River Delta, Southeast and Mekong River Delta and constitutes a share of 66.8%; 29.1% and 24.9%, respectively [30]. Depending on each local authority, the private sector will be eligible for incentive policies of water supply works. Overall, PSP in investment, management, and operation of water supply facilities in rural areas leads to greater operational efficiency and reduced rates of water loss in water supply facilities of Vietnam [28].

Customer satisfaction considerably affects the financial performance of water supply enterprises [31]. Many organizations and state agencies use customer satisfaction as one of the aspects to comprehensively evaluate the performance of water supply systems [32]. Customer satisfaction is one of the indicators contributing to the performance evaluation of water supply works [33]. Customer satisfaction significantly affects the business efficiency of water supply systems [34], impacting the management, operation, and maintenance of rural water supply schemes [35]. The factors associated with the characteristics of the

water supply work affecting customer satisfaction are the length of the pipeline, number of households [36], and lifespan of work [37]. Moreover, the level of user satisfaction of water supply systems depends on different factors such as water availability time, water pressure, water quality [38,39], or distance from the pump station to household [40]. Water prices and customer satisfaction are inter-related in a negative relationship [41,42]. However, no relationship was found between customer satisfaction and economic characteristics [40,43].

To compare the customer satisfaction of two water supply works corresponding to two different management organizations, it is necessary to consider the characteristics of the two works; this means that the two water supply works must be comparable in features. Therefore, an assessment model is needed to measure customer satisfaction of water supply systems. Some essential criteria such as water quality, the quantity of supplied water, and company responsibilities are used in the proposed model [38]. Decentralized techniques are used to integrate all these indicators in one unit indicator. The proposal model combines customer opinion into one unit to measure customer satisfaction. In the states of the Gaza Strip [39], the results of this study show that most of the respondents were dissatisfied with water services due to the amount of water, quantity, and continuity of water. Variables related to the system's features affecting the satisfaction are the pipeline length and the number of households by design [36]. Customer satisfaction is not related to economic, social, and demographic characteristics [40,43]. Service life affects customer satisfaction, often considered when designing a construction [37]. Water prices and customer satisfaction correlate, and it is an inverse relationship [41,42]. Water supply systems are classified as a natural monopoly due to the particularities of the operation and the required technical requirements [44]. It is unrealistic to build parallel water supply systems in the same area because investment costs are very high. It is challenging to have competition in the management and maintenance of water facilities [45]. Therefore, to assess the sustainability of a water supply scheme managed by an organization, it is necessary to incorporate an additional important factor, customer satisfaction, as a comparison method.

The private sector participation is significantly increasing in water supply projects, especially in rural areas, with management qualifications, and water supply systems are often small and medium-size. It is necessary to consider the customer satisfaction of the private sector compared with other sectors to assess service quality and improve the service quality of water supply in the private sector. Findings from this study may provide helpful information for the public sector, private sector, and other sectors to consider appropriate criteria to improve organization and management in the future. The remainder of this paper is organized as follows: Section 2 shows a research model used to evaluate customer satisfaction with rural water supply projects managed by different organizations; Section 3 describes data collection, including the characteristics of the data collection location and the water users; Section 4 explains the features of different water supply groups; Section 5 details the study results and discussion will be presented. The paper ends with some conclusions in Section 6.

## 2. Research Model

The Ordered Logit regression model is used to evaluate the satisfaction of water users with rural water supply projects managed by different organizations. Many studies used this model to assess customer satisfaction when the dependent variable has an ordinal like the Likert scale [22,26,46,47]. Some case studies have used this model to evaluate customer satisfaction [48,49]. This paper uses the Likert scale to assess water user satisfaction, which is appropriate and commonly used in social and behavioral science research [50,51].

Respondents rank their satisfaction with the rural water supply system from 1 to 5 (1 mean "very dissatisfied," 5 mean "very satisfied"). Let $y_i$ be individual i's response to the survey question, and assume that this can take one of the integer values 1, 2, 3, . . . , j. Let $y_i^*$ $-\infty < y_i^* < +\infty$ be the underlying latent variable representing respondent i's propensity to

agree with the statement advanced. Ordered Probit models are based on the assumption that $y_i*$depends linearly on $x_i$, according to the following [52]:

$$y_i* = x_i\beta + \varepsilon_i \tag{1}$$

in which

$y_i*$: is latent variable or customer satisfaction.

$x_i$: is a vector of independent variables, there are 12 independent variables described in Table 1.

**Table 1.** Independent variables in the model.

| No | Variable | Description | Symbol | Sources |
|----|----------|-------------|--------|---------|
| 1 | Gender | Respondent (1–male; 0–female) | $X_1$ | [26,48,53] |
| 2 | Age | Age of respondent | $X_2$ | [26,48,53] |
| 3 | Educational level | Respondent's qualifications (1–Unschooling, 2–Below elementary school, 3–Under secondary school, 4–Below high school, 5–Professional high school/college, 6–Undergraduate, 7–Post graduate) | $X_3$ | [26] |
| 4 | Income | Income of respondents (1–Below 2 million VND, 2–From 2 to 5 million VND, 3–From 5 to 10 million VND, 4–Above 10 million VND) | $X_4$ | [26] |
| 5 | Color of water | Ranking 5 Likert scale (1–Very Bad, 2–Bad, 3–Normal, 4–Good, 5–Very Good) | $X_5$ | [38,39] |
| 6 | Smell of water | Ranking 5 Likert scale (1–Very bad, 2–Bad, 3–Normal, 4–Good, 5–Very good) | $X_6$ | [38,39] |
| 7 | Taste of water | Ranking 5 scale (1–Salty, 2–Brackish, 3–Sour, 4–Good, 5–Very good) | $X_7$ | [38,39] |
| 8 | Available water time | Time available for water supply schemes (1–24h, 2–day only, 3–night only, 4–Half a day, 5–days with days or not, 6–times sometimes or not) | $X_8$ | [33,38,39] |
| 9 | Lifespan | Lifetime of schemes (years) | $X_9$ | [37] |
| 10 | Design capacity of the project | Design capacity of water supply schemes in rural areas ($m^3$/day–night) | $X_{10}$ | [36] |
| 11 | Water price | Water price of water supply schemes (VND/$m^3$) | $X_{11}$ | [26,41,42,48,53,54] |
| 12 | Management sectors | Management areas included (1–Private sector; 2–Public sector; 3–Other sectors) | $X_{12}$ | [26] |

Source: Summarized by the authors.

$\varepsilon_i$: is random error.

i = 1,2, ... n is the $i^{th}$ respondent, where n is the sample size (This study n = 1200).

$\beta$: is a vector of parameters that do not contain an intercept. Those parameters are considered as parameter slope in linear regression.

Respondents' choice is linked to latent variable with five options in the study as follows:

$y_i = 1$ (very dissatisfied) if $-\infty \le y_i* < a_1$

$y_i = 2$ (not satisfied) if $a_1 \le y_i* < a_2$

$y_i = 3$ (normal) if $a_2 \le y_i* < a_3$

$y_i = 4$ (satisfied) if $a_3 \le y_i* < a_4$

$y_i = 5$ (very satisfied) if $a_4 \le y_i* < +\infty$

The parameter $a_m$, with m = 1,2,3,4 and where $a_1 < a_2 < a_3 < a_4$ mean that four parameters are ranked hierarchically, and separated by cutoffs or threshold parameters. That is, we observe an individual $y_i$ in five hierarchical classifications that are separated by threshold parameters or cutoffs, i.e., coefficients a. In other words, threshold parameters demarcate the limits of different classifiers.

In this study, the authors use two models to examine the influence of different factors on customer satisfaction with rural water supply projects. In particular, the first model

considers the management sector, interviewee characteristics, water quality, and availability of water. In comparison, the second model includes all factors of model 1 and adds more features of water supply systems, including the design capacity and lifespan of the system.

## 3. Data Collection

### 3.1. Characteristics of Location

In order to ensure the representativeness of management organizations and users of the water supply system from beginning to end of the system, this research uses a stratified sampling method according to three groups of water users included at the top, the middle, and the bottom of the system. In 2015, a survey was conducted with 30 systems in three provinces, namely Thai Binh, Ha Nam, and Long An. Each system was randomly selected to survey 30 households from the beginning to the bottom each system. In 2018, Nghe An province (central region) selected two rural water supply systems; each scheme selected 150 households using water consisting of a beginning, middle, and end of the system with 50 households of each location.

The questionnaire survey to evaluate the level of satisfaction of rural water users includes 35 questions divided into three parts: (1) availability and access to water resources; (2) reliability of water supply service; and (3) water quality.

The questionnaire survey was funded by the Australian Government Department of Foreign Affairs and Trade (DFAT). Four organizations participated in collecting data including Institute for Sustainable Futures at the University of Technology Sydney (ISF-UTS), Institute for Water Resources Economics and Management (IWEM), East Meets West Foundation (EMWF), and the Center for Environmental and Natural Resources Research, Vietnam National University, Hanoi (CRES-VNU). This research updated missing information, including (1) Age, (2) Education level, (3) Occupation, (4) Income per capital, and (5) Water price. The authors conducted a telephone survey with 900 water users belonging to 30 rural water supply schemes in Ha Nam, Thai Binh, and Long An province in 2018. At the same time, a survey (see Table 2) collected 300 households using water from rural water supply schemes in Nghe An province (Central region), including 150 households using water managed by the community (other sectors), and 150 households using water managed by Provincial Centre for Rural Water Supply and Environmental Sanitation (the public sector).

**Table 2.** Number of households selected for the survey by three regions of Vietnam.

| Region/ Province | Number of Schemes | | | Number of Households | | | Collection Time |
|---|---|---|---|---|---|---|---|
| | Private Sector | Public Sector | Other Sectors | Private Sector | Public Sector | Other Sectors | |
| **Northern** | **10** | **2** | **8** | **330** | **70** | **200** | |
| Ha Nam | 5 | 1 | 4 | 150 | 30 | 120 | 2015, 2018 |
| Thai Binh | 5 | 1 | 4 | 180 | 40 | 80 | |
| **Central** | **0** | **1** | **1** | **0** | **150** | **150** | |
| Nghe An | | 1 | 1 | | 150 | 150 | 2018 |
| **South** | **5** | **3** | **2** | **150** | **90** | **60** | |
| Long An | 5 | 3 | 2 | 150 | 90 | 60 | 2015, 2018 |
| Total | **15** | **6** | **11** | **480** | **310** | **410** | |
| | | **32** | | | **1200** | | |

Source: Area survey of authors.

### 3.2. User Characteristics

Collected survey data shows that 54% of the interviewees are male, of which the age group 25-63 accounts for 85%. The education level of the interviewees is mainly below high school (see Table 3).

**Table 3.** Descriptive statistics of households using water.

| No | Interviewee Characteristics | Number of Households | Ratio (%) |
|----|----------------------------|---------------------|-----------|
| 1 | **Sex** | | |
| | Male | 774 | 64.50 |
| | Female | 426 | 35.50 |
| 2 | **Range of age** | | |
| | Under 35 | 14 | 1.17 |
| | From 35 to < 45 | 241 | 20.08 |
| | From 45 to < 55 | 393 | 32.75 |
| | From 45 to < 65 | 391 | 32.58 |
| | Above 65 | 161 | 13.42 |
| 3 | **Educational level** | | |
| | Unschooling | 29 | 2.42 |
| | Under elementary school | 142 | 11.83 |
| | Under secondary school | 576 | 48.00 |
| | Under high school | 359 | 29.92 |
| | Professional high school/college | 78 | 6.50 |
| | Undergraduate | 10 | 0.83 |
| | Post-graduate | 6 | 0.50 |
| | **Total** | **1200** | **100** |

Source: Author's collection results.

## 4. Characteristics of Private, Public, and Community Groups

In Vietnam, there are currently 16,573 projects, of which around 1579 works are managed by the private sector, accounting for 9.5%. The private sector participation is concentrated in lowland areas with higher population density and income than the other regions. The Red River Delta area has the most significant number of projects participating in the private sector, accounting for 66.8% [52]. As shown in Table 4 below, most rural water supply systems are well managed; the total of rate of sustained and medium status accounts for 68.4%. Sustained status works are mainly located in the Red River Delta and Mekong River Delta with 60% and 65.1%.

The increasing requirements for water quality make it difficult to ensure sustainable maintenance of the achieved results. Vietnam currently has 51% of the rural population using water that meets the standards by the end of 2020. Over 31 million people in rural areas account for 49% of the rural population without access to hygienic water. The percentage of people using hygienic water from concentrated water supply works in many localities is still low, and there is a large disparity between different regions. As shown in Table 4, the current water supply works operate inefficiently, and the inactive status is relatively high (31.6%), leading to a lack of sustainability in water supply activities. In addition, with the extreme impacts of global climate change and the increasing demand for socio-economic development, water resources are reduced in quantity and quality, especially in areas having frequent drought, water shortage, saltwater intrusion, and water pollution. The future demand for water supply systems for households in rural areas is increasing. The quality of water supply services is a multi-dimensional value, which means when providing a better-quality service, the more resources and costs are required. Therefore, the different water prices can be explained because the service quality of water supply systems is not the same [55].

By 2021, in Vietnam, the average price of water from rural areas for domestic use is approximately 8,000 VND/m$^3$. Specific water prices of water supply works are issued by each locality but are based on the price bracket of the Ministry of Finance. Water prices in rural areas are applied at roughly the same rates as in urban areas, but some localities have higher water prices in rural areas than in urban areas due to population density, higher investment, and management costs.

**Table 4.** Current status of systems by 2020.

| Subgroups | | | Area | | | | | | Total |
|---|---|---|---|---|---|---|---|---|---|
| | | | Northern Mountain | Red River Delta | North Central | South Central | Highlands | South East | |
| Total | | N | 8655 | 801 | 1364 | 1326 | 1300 | 316 | 16,573 |
| | | % | 100 | 100 | 100 | 100 | 100 | 100 | 100 |
| Management Organization | Community | N | 6,652 | 29 | 572 | 119 | 832 | 52 | 8,340 |
| | | % | 76.86 | 3.62 | 41.94 | 8.97 | 64.00 | 16.46 | 50.32 |
| | Cooperative | N | 111 | 66 | 16 | 42 | 12 | 27 | 300 |
| | | % | 1.28 | 8.24 | 1.17 | 3.17 | 0.92 | 8.54 | 1.81 |
| | Private sector | N | 156 | 535 | 38 | 52 | 7 | 92 | 1579 |
| | | % | 1.80 | 66.79 | 2.79 | 3.92 | 0.54 | 29.11 | 9.53 |
| | Commune People's Committee | N | 1675 | 126 | 712 | 1006 | 325 | 24 | 4785 |
| | | % | 19.35 | 15.73 | 52.20 | 75.87 | 25.00 | 7.59 | 28.87 |
| | Provincial Centre for Rural Water Supply and Environmental Sanitation | N | 61 | 45 | 26 | 107 | 124 | 121 | 1,569 |
| | | % | 0.70 | 5.62 | 1.91 | 8.07 | 9.54 | 38.29 | 9.47 |
| Current Status | Sustained | N | 2151 | 481 | 301 | 326 | 276 | 124 | 5489 |
| | | % | 24.85 | 60.05 | 22.07 | 24.59 | 21.23 | 39.24 | 33.12 |
| | Medium | N | 3473 | 185 | 585 | 411 | 396 | 116 | 5847 |
| | | % | 40.13 | 23.10 | 42.89 | 31.00 | 30.46 | 36.71 | 35.28 |
| | Ineffective | N | 1698 | 52 | 290 | 332 | 221 | 44 | 2814 |
| | | % | 19.62 | 6.49 | 21.26 | 25.04 | 17.00 | 13.92 | 16.98 |
| | Inactive | N | 1333 | 83 | 188 | 257 | 407 | 32 | 2423 |
| | | % | 15.40 | 10.36 | 13.78 | 19.38 | 31.31 | 10.13 | 14.62 |

Source: General Department of Water Resources (2021); N: the number of water supply systems.

The private sector's water supply facilities are more extensive than those managed by the public sector and other sectors. The capacity of the facilities managed by the private sector and public sector are medium, and large-scale projects with a service capacity of over 2000 m$^3$/day–night, about over 2000 households. However, facilities managed by the private sector are more diverse than those managed by the public sector; for example, water supply schemes under public sector management are at least 640 m$^3$/day–night, while those managed by the private sector are 300 m$^3$/day. However, the design capacity of the largest project managed by the private sector is three times higher than that of the public sector and five times higher than that of other sectors. Meanwhile, water supply works managed by other sectors mainly have a small capacity, simple treatment system, with a minimum capacity of 100 m$^3$/day–night. Most of the schemes managed by the private and public sectors are intercommune and interdistrict projects. Meanwhile, the schemes managed by other sectors are concentrated mainly in one village or one commune (see Table 5).

**Table 5.** Design capacity classified by management sectors.

| Indicator | Private Sector (m$^3$/Day–Night) | Public Sector (m$^3$/Day–Night) | Other Sectors (m$^3$/Day–Night) |
|---|---|---|---|
| Mean | 2548.94 | 2322.10 | 802.14 |
| Maximum | 10,500.00 | 3515.00 | 1720.00 |
| Minimum | 300.00 | 640.00 | 100.00 |
| Standard Deviation | 2934.75 | 1204.60 | 709.16 |

Source: Calculated by the authors.

## 5. Research Results and Discussion

### 5.1. Data and Description of Variables

Among the total number of interviewees, the number of households using water supply systems managed by the private sector is the highest, followed by the other sector, and

the public sector with 480, 410, and 310 respondents, respectively (see Table 2). The private sector's level of satisfaction (including satisfied and very satisfied) accounts for the highest rate with 62.98%, followed by the public sector accounting for around 60.32%, and other sectors represent approximately 40.95% (see Table 6).

**Table 6.** Satisfaction of water users by different sectors.

| Sector | Very Dissatisfied | Dissatisfied | Normal | Satisfied | Very Satisfied | Total |
|---|---|---|---|---|---|---|
| Private sector | 0.85% | 6.60% | 29.57% | 52.55% | 10.43% | 100% |
| Public sector | 3.23% | 7.10% | 29.35% | 49.03% | 11.29% | 100% |
| Other sectors | 6.90% | 14.05% | 38.10% | 37.14% | 3.81% | 100% |
| Total | 3.58% | 9.33% | 32.50% | 46.26% | 8.33% | 100% |

Source: Calculated by the authors.

As can be seen in Table 7, the proportion of men interviewed accounted for 64.50%; the average age was 53.21 years, and the education level was mainly below high school (about 90% of total interviewees). The household's average income is about 2–5 million VND/month, accounting for approximately 50%.

**Table 7.** Characteristics of water users and rural water supply facilities.

| Variable | Variable in Model | Mean | Standard Deviation | Min | Max |
|---|---|---|---|---|---|
| $X_1$ | Gender (1–Male, 0–Female) | 0.645 | 0.479 | 0 | 1 |
| $X_2$ | Age (year old) | 53.213 | 10.398 | 26 | 90 |
| $X_3$ | Educational level (1–Unschooling, 2–Under elementary school, 3–Under middle school, 4– Under high school, 5–Professional high school/college, 6–Undergraduate, 7–Postgraduate) | 3.308 | 0.914 | 1 | 7 |
| $X_4$ | Income (1–Below million VND, 2–From 2 to 5 million VND, 3–From 5 to 10 million VND, 4–Above 10 million VND) | 2.283 | 0.720 | 1 | 4 |
| $X_5$ | Color of water (1–Very bad, 2–Bad, 3–OK, 4–Good, 5–Very good) | 3.635 | 0.655 | 1 | 5 |
| $X_6$ | Smell of water (1–Very bad, 2–Bad, 3–OK,4–Good, 5–Very good) | 3.400 | 0.675 | 2 | 5 |
| $X_7$ | Taste-water (1–Salty, 2–Brackish, 3–Sour, 4–Good, 5–Very Good) | 4.221 | 0.818 | 2 | 5 |
| $X_8$ | Time (1–24 h, 2–day only, 3–nightonly, 4–Half a day, 5–days with daysor not, 6–times sometimes or not) | 3.358 | 1.864 | 1 | 6 |
| $X_9$ | Lifespan (year) | 9.502 | 6.675 | 2 | 30 |
| $X_{10}$ | Design capacity of the project | 1876.227 | 2134.171 | 100 | 10,500 |
| $X_{11}$ | Water price (VND/m$^3$) | 6640.667 | 1185.273 | 3000 | 10,000 |
| $X_{12}$ | Management sectors (1–Private sector, 2–Public sector, 3–Other sectors) | 1.958 | 0.861 | 1 | 3 |
| Y | Satisfaction (1–Very dissatisfied, 2–Dissatisfied, 3–Normal, 4–Satisfied, 5–Very satisfied) | 3.464 | 0.904 | 1 | 5 |

Source: Calculated by the authors.

For water quality, according to users, the color of water rated as "good" accounted for 67.39%, compared to the smell of water, approximately 90% of water users rated "good" and "very good". Regarding water taste, water users rated approximately 85% of water taste

as "good" and "very good". For the amount of water, the total rate of "24 h", "day only", "night only", "half a day" is 68%, which shows that the availability of water is not low.

Regarding the characteristics of the rural water supply system, the average lifespan of schemes in 2018 is 9.5 years, including a minimum of two years and a maximum of 30 years. This number indicates that most water supply systems have stabilized operation and management. The average design capacity of the project is 1876 households; the minor scale is 100 households, and the largest one is in 10,500 households.

The average water price is 6641 VND/m$^3$ approximately 0.263 EURO per m$^3$; minimum is 0.119 EURO/m$^3$, and the maximum is 0.397 EURO/m$^3$ (the currency exchange rates 1 EURO = 25,242.76 VND, 10 March 2022 of Vietcombank). Water price is equal to 0.119% of GDP per capita of rural people in Vietnam in 2021 (General Statistics Office of Vietnam). The proportion of surveyed water supply schemes managed by the private sector accounted for 39%, while the public sector accounted for 26%, and other sectors managed 35% (see Table 7).

### 5.2. Results and Discussion

As mentioned above, this research will analyze two models to choose the most suitable model. Model 1 included factors affecting water user satisfaction such as user characteristics (gender, education level, income), water quality (color, odor, and taste), duration of water availability, and economic factors (administration sector, water price).

With model 2, we keep the variables of model 1 and adding variables related to the characteristics of the building, such as size and lifespan. This model compares the characteristics of water supply projects in rural areas through the scale of the schemes (designed capacity) and lifespan. Water supply works have exclusive characteristics, significant investment costs, and specific technical requirements [44]; building two parallel water supply systems in one area is unrealistic [45] (see Table 8).

Model 1 with Chi-Square value resulted in 620.32 and was statistically significant. Therefore, the assumption was rejected because all regression coefficients of independent variables are 0 (Prob > chi2 = 0.000) with Pseudo $R_2$ of 0.2037. With economic factors, water-user satisfaction in systems is managed by the private sector is higher than that of other sectors. The higher the water price, the lower the customer satisfaction. Water prices in the private sector are higher than public sector [28,36,56] and other sectors [28].

Respondent's characteristics, including age, gender, and education level, do not affect customer satisfaction; this result is similar to previous research [26,48,53]. The higher the income of the interviewee, the lower the water-user satisfaction, and the results are similar to a previous study [26]. The better the water quality, the higher user satisfaction. The smell of water has the most significant influence on customer satisfaction, and the color of water does not affect customer satisfaction much. Moreover, the time availability of water influences customer satisfaction; similar to previous research [33,39], the longer water is available, the higher the customer satisfaction.

Model 2 has Pseudo R-squared and LR Chi-squared equal to 634.64 larger than model 1; therefore, model 2 is more appropriate and explains factors affecting user satisfaction better than model 1. The influencing factors include economic factors, interviewee characteristics, water quality, and water availability time, similar to model 1. However, model 2 depends on other variables such as system characteristics, lifespan, and design capacity schemes.

The larger the scale of rural water supply work, the lower the satisfaction of water users. Small-scale systems with a capacity of < 500 m$^3$/day/night are mainly managed by the community, Commune People's Committee, or cooperative. This result is similar to the findings of other studies [37].

According to the results, interviewee features do not affect customer satisfaction. Comparing the schemes managed by other sectors and the private sector, the level of "satisfied" and "very satisfied" increased by 4.76% and 2.41%, respectively. In contrast, for "undecided", "dissatisfied", and "very dissatisfied", water users in other sectors are higher than in the private sector by 3.63%, 2.29%, and 1.25%, respectively (see Table 9).

**Table 8.** Ordered Logit model results on water user satisfaction.

| Variables | Model 1 Coefficient (Standard Error) | Model 2 Coefficient (Standard Error) |
|---|---|---|
| **Economic factor** | | |
| Management area | | |
| Public sector | −0.0582 (0.160) | −0.158 (0.163) |
| Other sectors | −0.420 *** (0.156) | −0.422 ** (0.165) |
| Price | −0.000147 *** (0.0000552) | −0.000166 *** (0.0000561) |
| **Respondent's characteristics** | | |
| Gender | −0.107 (0.126) | −0.0974 (0.127) |
| Age | −0.00275 (0.00600) | −0.00291 (0.00602) |
| Education level | | |
| Below level 1 | 0.200 (0.405) | 0.203 (0.408) |
| Below level 2 | 0.379 (0.381) | 0.375 (0.384) |
| Below level 3 | 0.470 (0.388) | 0.432 (0.392) |
| Professional high school/college | 0.526 (0.439) | 0.501 (0.442) |
| Undergraduate | 0.215 (0.731) | 0.204 (0.733) |
| Postgraduate | 0.683 (0.863) | 0.608 (0.863) |
| Income | | |
| From 2 to 5 million VND | −0.492 *** (0.190) | −0.456 ** (0.192) |
| From 5 to 10 million VND | −0.417 ** (0.207) | −0.391 * (0.209) |
| Over 10 million VND | -0.221 (0.357) | −0.213 (0.356) |
| **Water quality** | | |
| *Color* | | |
| Bad | −1.083 (1.617) | −1.150 (1.601) |
| OK | −0.902 (1.619) | −0.941 (1.602) |
| Good | −0.0436 (1.617) | −0.0972 (1.601) |
| Very good | 1.799 (1.709) | 1.590 (1.695) |
| *Smell* | | |
| OK | 0.656 *** (0.245) | 0.679 *** (0.245) |
| Good | 1.578 *** (0.264) | 1.587 *** (0.264) |
| Very good | 3.270 *** (0.434) | 3.294 *** (0.436) |
| *Taste* | | |
| Sour | 0.394 (0.325) | 0.291 (0.327) |
| Good | 1.429 *** (0.322) | 1.366 *** (0.325) |
| Very good | 1.280 *** (0.323) | 1.190 *** (0.330) |
| **Time available for water** | | |
| Day only | −0.371 ** (0.184) | −0.593 *** (0.195) |
| Night only | −1.559 *** (0.354) | −1.693 *** (0.357) |
| Half a day | −1.078 *** (0.202) | −1.295 *** (0.212) |
| Days with days or not | −2.354 *** (0.236) | −2.461 *** (0.241) |
| Times sometimes or not | −1.050 *** (0.196) | −1.189 *** (0.201) |
| **Characteristics of rural water supply schemes** | | |
| Lifespan | | −0.0413 *** (0.0112) |
| Design capacity of project (m3/day/night) | | −0.0000788 ** (0.0000358) |
| Constant cut1 | −4.614 *** (1.722) | −5.575 *** (1.727) |
| Constant cut2 | −3.009 * (1.719) | −3.965 ** (1.723) |
| Constant cut3 | −0.559 (1.717) | −1.493 (1.720) |
| Constant cut4 | 3.026 * (1.719) | 2.128 (1.719) |
| Observations | 1.199 | 1.199 |
| Log likelihood | −1212.26 | −1205.1022 |
| Pseudo R squared | 0.2037 | 0.2084 |
| LR Chi-squared (*p*-value) | 620.32 (0.000) | 634.64 (0.000) |

Note: Standard errors in parentheses *** $p < 0.01$, ** $p < 0.05$, * $p < 0.1$. Source: Calculated by the authors.

**Table 9.** Marginal effects for Model 2.

| Variables | Very Dissatisfied | Dissatisfied | Normal | Satisfied | Very Satisfied |
|---|---|---|---|---|---|
| Economic factors Management sector | | | | | |
| Public sector | 0.00424 | 0.00815 | 0.0143 | −0.0168 | −0.00980 |
| | (0.00448) | (0.00846) | (0.0147) | (0.0176) | (0.0100) |
| Other sectors | 0.0125 ** | 0.0229 ** | 0.0363 ** | −0.0476 ** | −0.0241 *** |
| | (0.00511) | (0.00919) | (0.0148) | (0.0195) | (0.00934) |
| Price | 0.00005 *** | 0.00009 *** | 0.00001 *** | −0.00002 *** | −0.00010 *** |
| | (0.00002) | (0.00003) | (0.00005) | (0.00006) | (0.00003) |
| **Respondent's characteristics** | | | | | |
| Gender | 0.00296 | 0.00528 | 0.00789 | −0.01040 | −0.00576 |
| | (0.00387) | (0.00688) | (0.01030) | (0.01350) | (0.00751) |
| Age | 0.00009 | 0.00016 | 0.00024 | −0.00031 | −0.00017 |
| | (0.00018) | (0.00033) | (0.00049) | (0.00064) | (0.00036) |
| Education level | | | | | |
| Below level 1 | −0.00759 | −0.0119 | −0.0144 | 0.0238 | 0.0102 |
| | (0.0160) | (0.0243) | (0.0279) | (0.0487) | (0.0195) |
| Below level 2 | −0.0132 | −0.0215 | −0.0278 | 0.0427 | 0.0198 |
| | (0.0153) | (0.0230) | (0.0260) | (0.0460) | (0.0182) |
| Below level 3 | −0.0149 | −0.0245 | −0.0324 | 0.0486 | 0.0233 |
| | (0.0155) | (0.0234) | (0.0269) | (0.0468) | (0.0189) |
| Professional high school/college | −0.0168 | −0.0281 | −0.0381 | 0.0555 | 0.0276 |
| | (0.0165) | (0.0257) | (0.0318) | (0.0508) | (0.0231) |
| Graduate | −0.00761 | −0.0120 | −0.0145 | 0.0239 | 0.0102 |
| | (0.0265) | (0.0425) | (0.0534) | (0.0845) | (0.0378) |
| Postgraduate | −0.0197 | −0.0336 | −0.0473 | 0.0659 | 0.0347 |
| | (0.0250) | (0.0448) | (0.0720) | (0.0855) | (0.0561) |
| Income | | | | | |
| 2–5 million VND | 0.0124 ** | 0.0234 ** | 0.0390 ** | −0.0454 ** | −0.0294 ** |
| | (0.00489) | (0.00948) | (0.0174) | (0.0179) | (0.0136) |
| 5–10 million VND | 0.0103 * | 0.0198 * | 0.0338 * | −0.0383 * | −0.0257 * |
| | (0.00531) | (0.0103) | (0.0187) | (0.0197) | (0.0145) |
| >10 million VND | 0.00523 | 0.0104 | 0.0189 | −0.0198 | −0.0147 |
| | (0.00916) | (0.0178) | (0.0312) | (0.0343) | (0.0239) |
| **Water quality** Color | | | | | |
| Bad | 0.0354 | 0.0697 | 0.103 | −0.153 | −0.0551 |
| | (0.0308) | (0.0743) | (0.177) | (0.171) | (0.110) |
| OK | 0.0263 | 0.0547 | 0.0892 | −0.122 | −0.0481 |
| | (0.0300) | (0.0737) | (0.177) | (0.170) | (0.110) |
| Good | 0.00187 | 0.00457 | 0.0107 | −0.0106 | −0.00651 |
| | (0.0295) | (0.0733) | (0.178) | (0.170) | (0.110) |
| Very good | −0.0155 | −0.0448 | −0.170 | 0.0580 | 0.172 |
| | (0.0295) | (0.0736) | (0.185) | (0.172) | (0.139) |
| Smell | | | | | |
| OK | −0.0296 ** | −0.0478 ** | −0.0483 *** | 0.106 *** | 0.0199 *** |
| | (0.0130) | (0.0191) | (0.0140) | (0.0390) | (0.00617) |
| Good | −0.0510 *** | −0.0952 *** | −0.144 *** | 0.220 *** | 0.0704 *** |
| | (0.0134) | (0.0203) | (0.0213) | (0.0424) | (0.0100) |
| Very good | −0.0642 *** | −0.136 *** | −0.316 *** | 0.235 *** | 0.281 *** |
| | (0.0139) | (0.0208) | (0.0354) | (0.0488) | (0.0589) |
| Taste | | | | | |
| Sour | −0.0160 | −0.0213 | −0.0118 | 0.0406 | 0.00848 |
| | (0.0191) | (0.0245) | (0.0115) | (0.0456) | (0.00901) |
| Good | −0.0523 *** | −0.0876 *** | −0.0971 *** | 0.178 *** | 0.0595 *** |
| | (0.0187) | (0.0250) | (0.0160) | (0.0458) | (0.0109) |
| Very good | −0.0483 ** | −0.0785 *** | −0.0798 *** | 0.158 *** | 0.0485 *** |
| | (0.0188) | (0.0252) | (0.0160) | (0.0466) | (0.0104) |

**Table 9.** *Cont.*

| Variables | Very Dissatisfied | Dissatisfied | Normal | Satisfied | Very Satisfied |
|---|---|---|---|---|---|
| **Time available for water** | | | | | |
| Day only | 0.00732 *** | 0.0209 *** | 0.0727 *** | −0.0537 *** | −0.0472 *** |
| | (0.00268) | (0.00712) | (0.0240) | (0.0182) | (0.0161) |
| Night only | 0.0365 *** | 0.0848 *** | 0.190 *** | −0.214 *** | −0.0974 *** |
| | (0.0138) | (0.0246) | (0.0334) | (0.0549) | (0.0169) |
| Half a day | 0.0228 *** | 0.0577 *** | 0.154 *** | −0.151 *** | −0.0836 *** |
| | (0.00505) | (0.0105) | (0.0264) | (0.0270) | (0.0152) |
| Days with days or not | 0.0781 *** | 0.147 *** | 0.222 *** | −0.333 *** | −0.115 *** |
| | (0.0141) | (0.0196) | (0.0232) | (0.0330) | (0.0148) |
| Times sometimes or not | 0.0198 *** | 0.0513 *** | 0.143 *** | −0.135 *** | −0.0792 *** |
| | (0.00458) | (0.00970) | (0.0248) | (0.0244) | (0.0150) |
| **Characteristics of rural water supply schemes** | | | | | |
| Lifespan | 0.00125 *** | 0.00224 *** | 0.00334 *** | −0.00439 *** | −0.00244 *** |
| | (0.000376) | (0.00063) | (0.00091) | (0.00121) | (0.000681) |

Note: Standard errors in parentheses *** $p < 0.01$, ** $p < 0.05$, * $p < 0.1$; Source: Author's calculation.

For water prices, price increases to 1000 VND/m$^3$, the probability that users rank "satisfied", and "very satisfied" will decrease by 6.00% and 3.00%. Currently, the Vietnamese Government has implemented water price compensation for the private sector participating in investment, management, and operation. Therefore, when transferring rural water schemes from other sectors to the private sector, the probability of increasing the level of "satisfied" and "very satisfied" is 7.17%.

As a result, household income is higher; satisfaction tends to decrease. In contrast, the income of water users increases from "less than 2 million VND" to "2–5 million VND"; "satisfied" and "very satisfied" decrease by 4.54%, and 2.94%, respectively. The percentage of interviewees having around 2–5 million VND income accounts for 50% of the total number of respondents; 2–5 million VND is the average income of households in Vietnam. However, customers' income is more than 10 million VND; satisfaction is not affected.

For water quality, the smell of water increased from "bad" to "very good", the level of "satisfied" increased to 23.50%, and the level of "very satisfied" increased to 28.10%. Taste of water affects customer satisfaction less than the smell of water; taste of water changes from "brackish" to "very good"; water user satisfaction increases to 15.8% (assuming these factors are constant).

For water time, this factor is changeable to customer satisfaction. When water availability is "24 h" to "night only", customer satisfaction reduces to 21.40% (assuming these factors are constant). With water availability from "24 h" to "days with days or not", the level of "satisfied" and "very satisfied" decreased to 33.30% and 11.50%, respectively. Thus, water availability has a direct and significant impact on customer satisfaction.

Regarding features of rural water supply system, if the lifespan of schemes increases by one year (assuming other factors remain constant), the level of "satisfied" decreases by 0.439%, and "very satisfied" decreases by 0.244%. Satisfaction levels decrease as the scale of the system increases. As a result, the capacity of managers to operate rural water supply facilities is still limited [29]. System capacity increases, and satisfaction decreases slightly; as the model, the capacity of the water supply system increases to 1 m$^3$/day–night, and the level of water user satisfaction decreases by 0.008%.

## 6. Conclusions

This study aims to compare the level of customer satisfaction in rural water supply facilities managed by the private sector, public sector, and other sectors. The authors analyzed 1200 households using water from 32 schemes from three regions of Vietnam (North, Central, South) in which Ha Nam and Thai Binh are two provinces in the North; Nghe

An is a province in the Central region, Long An is a province in the South. Water facilities managed by the private sector have higher water user satisfaction than other sectors (Communities, Cooperatives, and Commune People's Committees). Comparing the schemes managed by other sectors and the private sector, water user satisfaction rated at "satisfied" and "very satisfied" by water supply systems managed by the private sector is 7.17% higher than the other sectors. On the other hand, the other sector has a rating of "dissatisfied" and "very dissatisfied", 3.54% higher than the private sector. There is no significant difference in customer satisfaction for projects managed by the private and public sectors. According to the results, interviewee features do not affect customer satisfaction.

In contrast, for "undecided, "dissatisfied", and "very dissatisfied", water users in other sectors are higher than in the private sector by 3.63%, 2.29%, and 1.25%, respectively. Characteristics of interviewees, including gender, age, and education level, do not affect customer satisfaction. The income of water users highly affects customer satisfaction, especially household income with 2–5 million VND/month. Among the factors of color, smell, and taste of water, the smell of water has the most significant influence on water user satisfaction. When the smell of water ranked at "good" or higher, customer satisfaction at "satisfied" and "very satisfied" increased by 29.04% and 51.60%, respectively. Water user satisfaction is quite sensitive to water availability. From continuous daytime to intermittent water availability, the level of "satisfied", and "very satisfied" decreased by 44.8%. In addition, as lifespan and water price increase, customer satisfaction decreases. The research results are a basis for the private sector to consider improving their service quality in rural water supply system factors that highly affect water user satisfaction, such as water quality, water availability. For state management organizations, it is necessary to support technical capacity of the private sector to improve their management capacity. It is advisable to encourage systems transfer from other sectors to the private sector to improve service quality. The limitation of this study is that the survey time to collect data has two different periods in 2015 and 2018; the homogeneity of data may slightly affect the analysis results. In this study, the data of selected water supply works belong to two provinces in the North (Ha Nam, Thai Binh), one province in the Central region (Nghe An), and one province in the South (Long An). Therefore, in the future, there should be extensive and more profound investigation research in the Central and Southern regions to balance the data range and have a detailed assessment.

**Author Contributions:** Conceptualization; methodology, soft-ware, N.T.A.; writing and revising paper, N.H.D; writing and editing, D.T.T All authors have read and agreed to the published version of the manuscript.

**Funding:** The questionnaire survey was funded by the Australian Government Department of Foreign Affairs and Trade (DFAT); the organizations involved in collecting data included the Institute for Sustainable Futures at the University of Technology Sydney (ISF-UTS), Institute for Water Resources Economics and Management (IWEM), East Meets West Foundation (EMWF) and the Center for Environmental and Natural Resources Research, Vietnam National University, Hanoi (CRES-VNU) in 2015. This research was updated missing information by authors in 2018. The Article Publishing Charges was paid by authors.

**Institutional Review Board Statement:** Not applicable.

**Informed Consent Statement:** Not applicable.

**Data Availability Statement:** Not applicable.

**Conflicts of Interest:** The authors declare no conflict of interest. The funders had no role in the design of the study; in the collection, analyses, or interpretation of data; in the writing of the manuscript, or in the decision to publish the results.

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
