# Peer review of "Privatization in Rural Water Supply and Customer Satisfaction: An Empirical Case Study in Vietnam"

_sustainability, doi:10.3390/su14095537_

Round 1

Reviewer 1 Report

The paper entitled “Privatization in Rural Water Supply and Customer Satisfaction: An Empirical Case Study in Vietnam” by Nguyen Tuan Anh et al could provide some information for the deep understanding on the private sector improving their service quality in rural water supply system. Topic is interesting and certainly in the scope of the Journal.

In this paper, the authors collected data in 4 regions of North, Central, and South Vietnam in 2015 and 2018. The paper summarizes 12 independent variables and uses Ordered Logit model to analyze the impact of different variables on customer satisfaction. The results show that, water facilities managed by the private sector have higher water user satisfaction than other sectors, and there is no significant difference in customer satisfaction for projects managed by the private and public sectors. Meanwhile, characteristics of water users, including gender, age, and education level, do not affect customer satisfaction. Income of water users, the smell of water and water availability highly affects customer satisfaction. The research results provide a basis for the private sector to improve the quality of its services in rural water systems. However, further revisions are needed before publication will be recommended.

  1. Major comments.

(1) The abstract does not match the conclusion of the article. For example, it is mentioned in line 3 of the abstract that the purpose of the paper is that “to identify the difference between water supply facilities managed by the private sector and the public sector”. However, in the conclusion of the paper, the purpose of the study is “a basis for the private sector to consider improving their service quality in rural water supply system factors”. I suggest summarizing the research purpose and significance again in the abstract.

(2) In the introduction, the third line of the third paragraph, “Measuring water user satisfaction is the most critical indicator to enhance the water quality of water supply systems”. What do you mean? how to improve water quality by measuring water user satisfaction?

(3) Please make sure that the content of the paper is consistent. For example, P8, taste of water is “sweet”. However, in the previous section and table 8, it is “very good”. Please change "sweet" to "very good". In addition, P10, 6 Conclusion, “The authors analyzed 1,200 households using water from 32 schemes in 3 regions…”. But in the abstract, “Research data is collected from 1200 households using water from rural water supply systems managed by different sectors in 4 provinces of Vietnam”.

  1. Specific comments

(1)P1, in the Abstract, “ in 4 provinces of Vietnam (Ha Nam, Thai Binh, Nghe An, and Ha Nam)”. Please change “and Ha Nam” to “and Long An”.

(2)P4, “4 Characteristics of private, public, and community groups”, line 8 of the first paragraph, “less than 100 m3/day – night”. But in Table 5, the minimum value for the other sectors is 100 m3/day – night.

(3)P6, 5.1 Data and description of variables, line 4 of the first paragraph, “the number of water users in the private sector is highest, … and the private sector accounts for the smallest”, Whether the private sector accounts for the most or the least? Please rewrite this sentence.

(4)P7, 5.2 Results and discussion, the first sentence of paragraph 3, What is the means of “Due to the features of water supply facilities, there is often a natural exclusivity due to the specificity of the water supply operation and the necessary technical requirements .” ? Please rewrite this sentence.

Author Response

We send the response to the comments of the commentators in the attached file.

Reviewer 2 Report

In the present version of the manuscript, it has a minor contribution, which is not expected to advance the state-of-the-art. Some comments are given as follows:

  1. The theoretical contributions may be improved.
  2. The bibliography may be improved, as there are 29 out of 32 references, i.e. over 90% older than 5 years. Also, there are 19 out of 32 references, i.e. over 59% older than 10 years.
  3. The structure of the manuscript is not mentioned in the introduction.
  4. The literature review in the introduction should review the literature, but not only write them there.
  5. Future research directions should be suggested in the conclusion.
  6. This version needs professional proofreading to address the numerous grammatical errors and writing mechanics.

Author Response

(The authors gave the same response as above.)

Reviewer 3 Report

although the topic is very interesting, the way it has been presented is quite poor

limit the statistical analysis (put everythin in an annex) and stick to the findings in terms of socioeconomic and other aspects

more discussion and interpretion of the replies/findings is absolutely necessary

the paper should be restructured and practically rewritten

see some indicative comments in the annotated manuscript attached

Author Response

Based on the comments of Reviewer 3, the authors have added and edited into the following attached file. The contents that have been edited by the author are highlighted in red color.

Round 2

Reviewer 2 Report

The responses to the Reviewer's comments have been checked. Generally speaking, the authors have presented a better revision and given some reasonable explanations.

This version needs professional proofreading to address the grammatical errors and writing mechanics.

Author Response

The authors edited mistakes in the attached file below

Reviewer 3 Report

the paper has been adequately improved
Reference No2 is wrong. See annotated manuscript
“Best Practices of PPP projects in the water services sector”, 3rd Int. Conf. “Management of International Business & Economics Systems” (MIBES 2008), 675-690, Larissa/Greece, 2008

Author Response

Based on the reviewer's comments, we have some grammar corrections as attached file
